# Recent Advances in the Heterologous Expression of Biosynthetic Gene Clusters for Marine Natural Products

**DOI:** 10.3390/md20060341

**Published:** 2022-05-24

**Authors:** Yushan Xu, Xinhua Du, Xionghui Yu, Qian Jiang, Kaiwen Zheng, Jinzhong Xu, Pinmei Wang

**Affiliations:** 1Ocean College, Zhejiang University, Zhoushan 316021, China; xuyushan@zju.edu.cn (Y.X.); duxinhua@zju.edu.cn (X.D.); xiaohuige@zju.edu.cn (X.Y.); jiangqian11@zju.edu.cn (Q.J.); kevin_zhengplus@zju.edu.cn (K.Z.); xujinzhong@zju.edu.cn (J.X.); 2State Key Laboratory of Motor Vehicle Biofuel Technology, Zhejiang University, Zhoushan 316021, China

**Keywords:** marine natural products, heterologous expression, heterologous hosts, biosynthetic gene clusters, genetic manipulation

## Abstract

Marine natural products (MNPs) are an important source of biologically active metabolites, particularly for therapeutic agent development after terrestrial plants and nonmarine microorganisms. Sequencing technologies have revealed that the number of biosynthetic gene clusters (BGCs) in marine microorganisms and the marine environment is much higher than expected. Unfortunately, the majority of them are silent or only weakly expressed under traditional laboratory culture conditions. Furthermore, the large proportion of marine microorganisms are either uncultivable or cannot be genetically manipulated. Efficient heterologous expression systems can activate cryptic BGCs and increase target compound yield, allowing researchers to explore more unknown MNPs. When developing heterologous expression of MNPs, it is critical to consider heterologous host selection as well as genetic manipulations for BGCs. In this review, we summarize current progress on the heterologous expression of MNPs as a reference for future research.

## 1. Introduction

Oceans cover 71% of the Earth’s surface and approximately 87% of the biosphere, and possess specific pH, temperature, pressure, oxygen content, light, and salinity greatly different from the land [1,2]. With the development of deep-sea exploration technology, spectroscopy, X-ray crystallography, and various separation techniques, marine natural products (MNPs) have attracted widespread concerns [3]. The complexity and the multidomain of the chemical structures endow many MNPs with specific bioactivity, allowing them to be the ideal candidates for new drug development. Secondary metabolites (SMs) of marine microorganisms have pharmaceutical value [4]. Marine microorganisms inhabit almost all ecological niches that are available, and those who survive in harsh conditions secrete SMs with novel chemical structures and unique physiological properties [2,5]. From 2018 to 2020, over 40 bioactive compounds derived from marine microorganisms were reported, and they were compiled in Ameen’s review [4]. Notably, in 2018, brentuximab vedotin derived from cyanobacteria *Symploca* sp. was approved by the United States Food and Drug Administration (US FDA) to treat Hodgkin’s lymphoma cancer (https://www.fda.gov/drugs/resources-information-approved-drugs/brentuximab-vedotin, accessed on 19 April 2022). However, research on the biodiversity of marine microorganisms and the bioactivity of their metabolites is still very limited, since only 0.1% of marine microorganisms have been explored or even less than 0.01% [6,7]. Furthermore, epifluorescence microscopy and rRNA sequencing for microbial community analysis have revealed that microbes that can be cultured in laboratory represent only a small fraction of the microbial community, and in most cases, are not the dominant population under natural conditions [8]. At the same time, due to the extremely low reproducibility and the small amounts of activated compounds extracted from marine microbial samples, it is difficult to obtain sufficient MNPs for the further analysis of biological activities and medicinal properties. In addition, total chemical synthesis is hard to realize economically, especially for the MNPs with complex structures [9], which also hinders the subsequent studies such as the verification and optimization of drug structures.

Marine microorganisms aggregate and sequentially arrange the genes involved in the biosynthesis of a certain MNP as biosynthetic gene clusters (BGCs) [10,11]. Today, high-throughput DNA sequencing technologies coupled with convenient genome mining and metabolomic analysis tools allow researchers to rapidly identify BGCs in microbial genomes [12]. This technical breakthrough provides new avenues for the mining, modification, and production of MNPs. On the basis of genomics, we can subsequently use molecular techniques to biosynthesize high titers of MNPs, rather than merely using traditional separation and purification techniques to obtain small amounts of chemical compounds [13]. Genome mining is used to speculate the potential functions of the metabolites guided by BGCs [14,15]. Modern bioinformatic and genetic tools can also efficiently eliminate known metabolites from complex metabolite mixtures, increasing the rate of discovery of new products [16]. For example, the most widely used bioinformatics tool, antiSMASH [17], can provide detailed information about BGCs and output the stereochemical data about the assumptive compound and the amino acid module that builds the compound. Specifically, an open access database designed for MNPs called CMNPD (comprehensive marine natural products database) [18] has been available for researchers to determine the accurate chemical structures of MNPs and assist in calculating their physicochemical and pharmacokinetic properties. More bioinformatics tools and databases for the genome mining of MNPs are included in Ambrosino’s review [19].

A major challenge to biosynthesize MNPs is the low expression or silence of MNP BGCs in laboratory. Researchers have developed several triggering strategies to activate cryptic BGCs. For culturable strains, the one strain many compounds (OSMAC) strategy excavates the potential of a specific microorganism to produce different natural products (NPs) by systematically varying culture parameters including medium composition, aeration rate, culture vessel, and the addition of enzyme inhibitors [20]. Moreover, screening strains with mutations in RNA polymerase or ribosomal proteins, as well as the deletion or induction of global transcriptional regulators, are genetic engineering techniques that have successfully triggered the expression of cryptic BGCs in microorganisms [11]. In addition, epigenetic regulation and chromatin-level remodeling approaches can precisely control the homologous or heterologous expression of BGCs by expressing pathway-specific activation genes or deleting pathway-specific repressor genes. Likewise, strong promoters can be added to BGCs to stimulate strains to produce specific SMs [21,22,23]. More concrete BGCs awakening strategies are summarized in Choi’s review [24].

However, these strategies are not amenable for all BGCs of MNPs. Above all, studies have shown that a large proportion of marine microorganisms are not culturable, which impedes the extraction and identification of bioactive metabolites from them. So apparently, the OSMAC strategy does not work in unculturable strains, and this strategy requires a relatively large experimental volume for strains with low production of MNPs. In addition, the supplementation of the medium with chromatin modifiers (such as histone deacetylase inhibitors) is not able to activate the cryptic BGCs in all species of fungi [25]. Furthermore, for the majority of BGCs, the external and/or internal signals that trigger their expression remain unknown. Even for the culturable marine microorganisms, they are inclined to show a small metabolic profile under the traditional conditions in laboratory. Otherwise, other strategies such as cocultivation and mixed fermentation should be considered to stimulate the production [26]. Moreover, a low metabolic rate and complex metabolic background may also lead to missed metabolites. To overcome these challenges, expressing BGCs in a heterologous host with a clear background of metabolites to obtain MNPs of high quality and quantity becomes a reliable alternative.

The expression of BGCs cloned from uncultured species and environment samples is expected to facilitate the discovery of new products and produce valuable compounds [27,28,29]. A heterologous expression platform for marine microorganisms and marine environment samples in the engineered chassis strains provides a feasible method for the efficient production of MNPs, especially for the exploration of fungal MNPs. Fungi can produce a large number of bioactive metabolites [30], but due to the lack of mature fermentation technology and genetic manipulation tools, fungal MNPs rely heavily on a universal heterologous expression platform. The heterologous expression of terrestrial fungal NPs has made some progress, but most of the heterologous expression of MNPs is still focused on marine bacteria. The heterologous expression of MNPs has the following four advantages. First, a successful heterologous expression allows the correct DNA ligation and efficient co-work of all essential gene fragments in BGCs which indicate the biosynthesis of MNPs. Second, heterologous expression can biosynthesize MNPs from unculturable marine microorganisms and environment samples. Third, heterologous expression relies on a suitable host, reducing the burden of developing novel genetic tools for different strains. Fourth, the library of biosynthetic genes can be continuously updated and optimized, not only for heterologous expression, but also for the genetic manipulation of cloned pathways, biosynthetic studies, or gene remodeling of BGCs [10]. So far, the heterologous expression of MNPs has made several remarkable advances in expressing BGCs from marine actinomycetes [31] and cyanobacteria [32], which exhibits promising prospects for extending this strategy into more marine microorganisms. Efficient heterologous expression first requires the selection and optimization of a suitable heterologous host. Next, the selected BGCs are cloned, assembled into dedicated vectors, and then transferred to heterologous chassis strains for expression. In the end, using methods such as metabolic engineering can mass-produce specific MNPs for clinical research and therapeutic use [10,21,33]. Metagenomics shows an extraordinary potential in the heterologous expression of BGCs for MNPs (Figure 1), since this strategy directly extracts the total community DNA from marine samples without the need for the isolation or culture of microbes [34]. The combination of metagenomics and efficient bioactivity screening has become an effective method to identify novel bioactivities from inseparable marine microorganisms [6,12]. A marine metagenomic library was constructed from tidal flat sediments by Fujita et al. [27]. Through a metal-binding compound detection test, the researchers relied on the metal ion indicator chrome azurol S (CAS) for functional screening. The siderophore BGC was then cloned and vibrioferrin was heterologously expressed in *Escherichia coli*.

In this article, we review the research progress on the heterologous expression of MNPs. The heterologous expression of MNPs can be roughly summarized into two main steps: (1) appropriate selections of the heterologous hosts; (2) genetic manipulations for BGCs. These steps exert profound impacts on the effects of heterologous expression. It is clear that the intactness and arrangement of BGCs determine the expression at the genetic level, while the cloning of BGCs usually larger than 10 kb is not so simple. Meanwhile, the endogenous metabolic pathways of the hosts can affect the structures of MNPs, and the mechanisms are difficult to demonstrate and vary from host to host. For example, one research group cloned *ber* BGC, which indicated the formation of berninamycins A and B, while they finally obtained berninamycins J and K in another host, which pointed to the unknown host-dependent enzymes in the host [35]. Considering the above problems, this review focuses on these two steps, as well as advances in the heterologous expression of MNPs from marine BGCs, which can provide a reference for new MNPs exploration.

## 2. Heterologous Hosts for MNPs

The regulatory factors of heterologous hosts and/or biosynthetic factors control the heterologous expression of NPs [36]. When selecting the heterologous hosts, the characters of BGCs must be considered, including the required substrates, as well as the genetic and physiological properties of the original strains. In general, the closer the host is to the original strain, the more likely the transcription factors (such as regulatory factors, promoters, and ribosomal binding sites) of exogenous BGCs in hosts will work [37,38], due to the similar codon usage patterns shared by the evolutionarily closed strains and therefore, the higher translation efficiency. Each amino acid can be encoded by different synonymous codons whose frequencies vary with different species and the synonymous codon replacement in heterologous hosts would influence the secondary and tertiary protein structures [39]. Moreover, in closely related heterologous hosts, the exchange of promoters can improve the level of expression and thus increase the yield of target compound [40]. Some model strains have been modified to adapt to heterologous expression by eliminating the BGCs of their own metabolites. Heterologous hosts are categorized into four groups in Table 1. *Streptomyces* are the major force for Gram-positive bacteria and the model strains in *Streptomyces* play an outstanding role in the biosynthesis of antimicrobial agents derived from actinomycetes. Gram-negative bacteria mainly include *E. coli* and cyanobacteria. *E. coli* is an ideal heterologous host for bacterial BGCs, since *E. coli* is a well-developed model species. The model strains of cyanobacteria are also robust hosts for the bacterial BGCs of MNPs, particularly for the medicinal agents from all kinds of cyanobacteria. Other two types of heterologous hosts for MNP BGCs include *Aspergillus* that assists in expressing fungal MNP BGCs and microalgae *Fistulifera solaris* for algal BGCs. Different heterologous hosts for the expression of MNP BGCs are summarized and categorized in Table 1.

### 2.1. Escherichia. coli

*E. coli* not only grows quickly, but also lacks endogenous secondary metabolic pathways and thus has little background interference from metabolites. At the same time, abundant multifunctional gene manipulation tools can be implemented in *E. coli*. To make *E. coli* compatible with more BGCs from different sources, researchers need to recode the target BGCs by replacing the native codons with synonymous codons frequently used in *E. coli* to eliminate the codon bias between the native strains and hosts [84]. *E. coli* has been successfully applied to elucidate the enzymatic machinery of multiple biosynthetic pathways and has provided a good platform for the heterologous expression of bacterial and fungal NPs [85,86]. Almost all types of NPs can attempt to be produced in *E. coli*, including macrolides, cyclic peptides, terpenes, and alkaloids [87]. In 2005, two research groups [60,88] completed the first genetics-based identification, transfer, and heterologous expression of the biosynthetic pathways from marine microbial symbionts. Researchers cloned and expressed the cyanobacterial symbiont *Prochloron* spp. DNA in *E. coli*, and successfully synthesized the metabolite patellamide with antitumor activity. Since the ocean has a special environment of high salinity, the marine halophilic bacteria become good biological sources for salinity-tolerant enzymes. Yu et al. [61] cloned a BGC of multidomain β-1,4-endoxylanase with a high molecular weight, *xylM_18_* from the halophilic marine bacterium *Marinimicrobium* sp. LS-A18 into the *pET28a* vector and heterologously expressed it in *E. coli* BL21 (DE3). The purified β-1,4-endoxylanase XylM_18_ has high activity between pH 6.0~10.0, and the best activity appears at pH 7.0. Its preferable activity is within the range of NaCl 0.2~25% (*w*/*v*), making XylM_18_ a suitable candidate enzyme for the biodegradation of xylan under high salt and alkali conditions. However, *E. coli* often cannot efficiently complete the functional folding of key fungal biosynthetic enzymes such as the P450 family enzymes when constructing the entire biosynthetic pathway rather than just expressing core enzymes [21]. Moreover, *E. coli*’s heterologous expression of PKS, NRPS, and PKS/NRPS hybrid biosynthetic pathways has been proved to be more difficult [40].

### 2.2. Cyanobacteria

Cyanobacterial MNPs are diverse in structure and bioactivity, and more than 1100 SMs have been identified from cyanobacteria, most of which are produced by four genera: *Hapalosiphon*, *lyngbya* (*Moorea*), *Microcystis,* and *Nostoc* [89]. Known or predicted cyanobacterial MNPs account for approximately 20% of marine-derived small molecules currently used in clinical trials as well as the treatment of cancer, neurological disorders, infectious diseases, anti-inflammatory, and UV protection [90,91,92]. The molecular structures of cyanobacterial MNPs differ from terrestrial and freshwater species [93]. Cyanobacterial MNPs consist of nitrogen-rich scaffolds with distinguished structural diversity and usually have halogenated, methylated, and oxidative modifications [94]. Cyanobacterial MNPs are typically produced by PKS, NRPS, or a hybrid PKS/NRPS pathway with multiple tailoring steps [95]. *Anabaena* sp. PCC 7120 is a filamentous freshwater cyanobacterial strain that has been used as a model organism to study cellular differentiation, hydrogen production, and nitrogen reduction [58]. It has not been reported that PCC 7120 produces its own bioactive metabolites from the NRPS/PKS clusters, which provides a clear and clean compound background for genetic manipulation. In addition, PCC 7120 can apply established genetic tools and recognize the promoters of BGCs from different marine cyanobacteria, which can be used as an ideal heterologous host to express cyanobacterial MNPs. Videau et al. [58] expressed the *ltx* gene from *Moorea producens* in PCC 7120 and successfully obtained a 13-fold titer of the cytotoxic lyngbyatoxin A. Taton et al. [57] used TAR cloning to clone the long cryptomaldamide BGC (28.7 kb) of *M. producens* JHB, which was then integrated into PCC 7120 and successfully expressed.

### 2.3. Actinomycetes

High-GC-content Gram-positive mycelium bacteria are the microbial source of approximately two-thirds of known antibiotics [96]. By sequencing the actinomycete genomes, researchers have identified numerous cryptic BGCs of SMs with unknown function that may provide novel antibiotics for clinical use [97]. *Streptomyces* are particularly well known for producing medicinal compounds with antimicrobial and antitumor activity [98], and *Streptomyces* can even be processed into insecticides [99] and herbicides [100]. *S. coelicolor* A3(2) is a model species for studying the genetics and biology of actinomycetes, and a large number of genetic tools can be applied in this strain. After knocking out the BGCs of its main SMs, a series of heterologous expression hosts such as *S. coelicolor* M1146, M1152, and M1154 were constructed [101]. Chen et al. [44] heterologously expressed the *kmy* cluster in *S. coelicolor* M1152 to generate kendomycin B, which had strong antibacterial activity and moderate tumor cytotoxicity. The BGC was derived from a marine rare actinomycetes *Verrucossppora* sp. SCSIO 07399. This strain has a poor sporulation ability and slow growth rate, which hinders the stable production of kendomycin B. However, heterologous expression has improved the efficiency of researchers to obtain this MNP. Rodriguez et al. [51] identified the BGC and pathway for the antibiotic nybomycin from *S. albus* subsp. *chlorinus* NRRL B-24108 by heterogeneously expressing it in *S. albus* De114, a host without endogenous BGC interference. Zhang et al. [55] chose to develop a chassis strain marine actinomycete *Salinispora tropical* CNB-4401, which served as an ideal heterologous host to express BGCs. The researchers replaced the three genes (*sal*A-C) essential for the synthesis of salinosporamide in *Salinispora tropical* CNB-440 by a knockout cassette containing the phage attachment site *attB* of *S. coelicolor* ΦC31, resulting in the mutant *S. tropica* CNB-4401, which simplifies the chemical background of the original strain *S. tropical* CNB-440. The researchers successfully integrated and expressed the thiolactomycin BGC in *S. tropica* CNB-4401, validating the effectiveness of this new heterologous host. Compared to *S. coelicolor* M1152, *S. tropica* CNB-4401 produces approximately a threefold higher titer of thiolactomycin. This is the first example of utilizing marine actinomycetes as heterologous hosts for the expression of BGCs, providing an effective platform for accelerating the discovery of new MNPs and engineered biosynthesis pathways.

### 2.4. Others

*Bacillus subtilis* is a typical host with a low GC content and a frequently used heterologous expression platform where multiple molecular tools are applied [102]. Li et al. [56] cloned a large amicoumacin BGC from marine isolate *Bacillus subtilis* 1779 and activated this BGC in the host *B. subtilis*. Two intermediates of amicoumacin were detected. Apart from *E. coli* and *Synechococcus*, *Myxococcus xanthus* is also a good Gram-negative strain for heterologous expression and an antifungal agent haliangicin from a marine myxobacterium was produced with high efficiency in *M*. *xanthus* [81].

Yeasts are single-cell model eukaryotes that have been used as heterologous hosts to express biosynthetic pathways from microbes [103]. *S. cerevisiae* shows a high affinity of codon usage and a certain tolerance to exogenous bacterial sequences [104]. In addition, yeasts produce few endogenous SMs, minimizing competition of exogenous BGCs with cognate SM pathways and thus simplifying the detection and purification of target compounds [105]. The heterologous expression of BGCs from terrestrial bacteria and fungi has been proved workable in yeasts, while there have been few successful examples of BGCs from marine microorganisms expressed in yeasts. Many filamentous fungi are potential hosts for heterologous expression of fungal BGCs. Since filamentous fungi have highly developed secondary metabolic systems, they contain functional PPTases essential for the biosynthesis of fungal NPs. Furthermore, filamentous fungi were shown to recognize and correctly splice introns in exogenous fungal mRNA sequences without the need to clone and purify intron-free cDNAs from BGCs [106]. In addition, for the heterologous expression of fungal BGCs, since filamentous fungi are closely related, *Aspergillus* usually shows a good compatibility in codon usage, which facilitates translation [107]. Among filamentous fungal hosts, *Aspergillus* species such as *Aspergillus nidulans*, *Aspergillus niger*, and *Aspergillus oryzae* have been developed as efficient platforms for heterologous expression after the knockout of background metabolic pathways. The researchers cloned the BGC of beauvericin from marine-derived *Fusarium proliferatum* LF061 into the host *A. nidulans* [82]. The host successfully yielded beauvericin and the production was hundreds of times higher by replacing strong promoters and optimizing fermentation conditions. Despite the rapid progress in heterologous expression of terrestrial fungal NPs, the genetic tools for heterologous expression and homologous manipulation of marine fungi are still scarce, and the methods for terrestrial heterologous expression need to be transferred to the field of marine fungi [108].

Microalgae are promising heterologous hosts for MNPs. Microalgae with chloroplasts serve as photosynthetic production platforms, meaning that microalgae can grow rapidly by converting CO_2_ into energy without the need of nutritious carbon sources. Using marine microalgae as host strains, several studies have heterologously expressed the genes encoding key enzymes in the biosynthetic pathways of terrestrial plants [109,110,111]. For the exploration of MNPs, Maeda [83] inserted a *cox* gene derived from a red alga into a genetically engineered microalgal host *Fistulifera solaris*. The *cox* gene encoded the biosynthesis of cyclooxygenase. This enzyme catalyzed the transformation of the rich C20 polyunsaturated fatty acids in *F. solaris* into prostaglandins with high production.

## 3. Genetic Manipulations for BGCs

### 3.1. BGC Cloning

Due to the large size, high repeatability, and high GC content of BGCs in many microorganisms, cloning BGCs is a challenging and critical step for heterologous expression [10]. With advances in genome sequencing, bioinformatics and molecular biology techniques, many different types of cloning methods have been developed for the heterologous expression of MNPs. Large BGCs can be reconstructed from multiple amplified PCR fragments. Researchers have implemented a variety of BGC cloning or DNA assembly methods, including in vivo cloning and in vitro assembly. In vitro assembly is classic and straightforward, but it is constricted by enzymes, and library-based methods need laborious screening. In vivo cloning methods are flexible, while random mutations in sequences and incorrect ligation of DNA fragments cannot be circumvented either. The methods used in cloning MNP BGCs, as well as their advantages and disadvantages are listed in Table 2.

In vivo cloning methods are used to reconstitute large microbial BGCs. Transformation-associated recombination (TAR) cloning [112] is a prevalent method for cloning large size BGCs of MNPs. It exploits the yeast’s natural in vivo homologous recombination (HR) activity. Once the site-specific sequences at both ends of the linearized TAR cloning vector match the target genome sites, the HR of gDNA fragments in yeast competent cells starts [113]. The reconstructed TAR cloning vector is then shuttled into *E. coli* for subsequent propagation, verification, and sequencing (Figure 2a). TAR cloning vectors can also carry factors necessary for transport and a stable presence in a heterologous host. A 67 kb NRPS BGC from the marine actinomycete *Saccharomonospora* sp. CNQ-490 was integrated and captured using the yeast–*E. coli*–*Streptomyces* shuttle vector *pCAP01* and activated in the model actinomycete *S. coelicolor* [48]. The lipopeptide antibiotic taromycin A was synthesized and expressed. λ/Red [114] and Red/ET [115] systems exhibit a strong ability of HR in *E. coli* by virtue of the efficient phage-recombinases from the λ phage, and these two methods are continuously improved to directly clone DNA segments from complex genomes of marine microorganisms. Phage recombinases also function in a serine integrase recombination assembly (SIRA) [116]. These enzymes catalyze and recognize *attP* and *attB* sites to generate new *attL* and *attR* sequence linkages (Figure 2b). In this way, DNA fragments can be spliced together in a specific order and orientation to form a self-replicating structure and selectable markers.

Cosmid and fosmid libraries [117,118] are the commonly used in vitro cloning methods for MNP BGCs, since the construction of an open reading frame library assists in the rapid screening for BGCs with bioactivity from marine metagenomes. Large DNA fragments are carried in cosmid or fosmid vectors, from which researchers segregate the clones containing useful BGCs. The restriction enzyme-mediated assembly is a conventional in vitro assembly method which heavily relies on type IIs restriction enzymes for the cutting and ligation of DNA fragments. Many studies of heterologous expression of marine-derived enzymes in *E. coli* have been achieved by this simple method. Bacterial artificial chromosome (BAC) [119] vectors can replicate in stable condition in *E. coli* and increase DNA yield, particularly suitable for cloning BGCs into bacterial hosts. The overlap extension PCR method [120] uses primers with complementary ends to assemble multiple overlapping DNA fragments obtained by standard PCR into a full-length double-stranded DNA (Figure 2c). The Gibson assembly [121] is another in vitro method for joining multiple DNA fragments with the help of 5′ exonuclease (Figure 2d). Wang [122] systematically and comprehensively reviewed the latest BGC cloning methods.

### 3.2. BGC Regulation

Although the heterologous expression of BGCs in model hosts is a robust way to increase the production of MNPs, it still faces many problems due to the difference between the host and native strains in metabolic regulation, precursor supply, auxiliary factors, and the self-resistance, such as the lack of regulation of gene expression, the difficulty of recruiting the premise modules, and the inefficient transcription and translation of BGCs [13,33]. Therefore, it is important to rationally design and regulate BGCs and heterologous hosts to improve the biosynthetic efficiency of MNPs. At the same time, the expression of known BGCs in heterologous hosts may lead to the generation of new derivatives, which may be the result of incompatible metabolic systems or nonpathway modification of compounds by endogenous enzymes [40]. Kim et al. [41] heterologously expressed the entire barbamide BGC in an engineered strain of *S. venezuelae* DHS 2001 in which the picromycin PKS BGC was deleted, and obtained 4-*O*-Demethylbarbamide, a new derivative of barbamide which lacks one *O*-methyl. The molluscicidal activity of this new derivative was several times higher than that of the original structure, which was also the first study on the successful heterologous expression of the marine cyanobacterial NRPS/PKS BGC in a terrestrial host. The selection of a suitable host combined with the stable expression of exogenous genes after genetic engineering in hosts provide an important fundamental for biosynthesis and molecular engineering to study new MNPs. At present, two regulation methods for BGCs have been applied to develop MNPs, including (1) activating the cryptic pathway with a strong promoter; and (2) supplementing and directing the transport of precursor molecules.

Promoters are critical for transcription, while endogenous promoters of BGCs are usually not compatible with heterologous hosts unless they are closely related enough to successfully initiate gene transcription and regulation [123]. Thus, the heterologous expression of BGCs requires the simultaneous high-level expression of multiple promoters and controllable initiation of MNP biosynthesis under appropriate circumstances. Currently, constitutive promoters from housekeeping genes and inducible promoters from major metabolic genes are often used to drive the expression of BGCs in commonly used heterologous expression systems [21]. Bauman et al. [49] discovered, cloned, reconstructed, and heterologously expressed a cryptohybrid phenoazine-type BGC (*spz*) from the marine actinomycetes *Streptomyces* sp. CNB-091. The researchers introduced three promoter cassettes to reconstitute the *spz* cluster. An engineered bidirectional promoter combining the strong synthetic promoters *p21* and *sp44* was introduced into the cluster to control the biosynthesis of the core genes; the *actI* promoter was used to control the PKS gene of the *spz* cluster; the *ermE** promoter was used to regulate the expression of the cytochrome *bd* oxidase gene. Overexpression of the *spz* cluster increased the yield and chemical diversity of phenazine MNPs, and the researchers detected at least 38 streptomycin phenazines from the fermentation liquor.

Some MNPs are produced in very low yields, and even heterologous expression systems using strong promoters may not necessarily produce sufficient amounts of the required compounds. This low yield severely limits the characterization of marine bioactive molecules and reduces their industrial production. This limitation may be circumvented by enhancing precursor supply to facilitate MNP biosynthesis [124]. Marine myxomycete *Haliangium ochraceum* SMP-2 produces the antifungal polyketide haliangicin, but its yield is low. BGC *hli* (47.8 kb), which encodes haliangicin, has been identified and expressed in *Myxococcus xanthus* to assure the efficient production of haliangicin. It showed that the titer of haliangicin increased significantly when the precursor concentration was greater than 50 mM. When the concentration reached 200 mM, haliangicin at a concentration of 11.0 ± 2.1 mg/L was detected after 5 d. Compared with the original strain, the yield increased 10 times and the growth rate increased 3 times [81].

Recently, CRISPR-Cas9 (clustered regularly interspaced short palindromic repeats) [125] has unleashed its great potential in genetic manipulation, especially showing tremendous advantages over traditional techniques in the efficient site-specific mutagenesis and in-frame deletions. By using CRISPR-Cas9, researchers are able to delete or replace the predicted key gene loci of connate BGCs in host strains, blocking these biosynthetic pathways, reducing the production of SMs and thus forcing more substrates to get involved in the biosynthesis of exogenous BGCs. Moreover, researchers can use CRISPR-Cas9-mediated knock-in strategy to precisely introduce promoter cassettes to replace original promoter regions, which might stimulate the high expression of inserted BGCs or produce new derivatives. Several experiments have succeeded in activating *Streptomyces* and cyanobacteria BGCs [126,127], and the CRISPR-Cas9 technique can possibly be extended to more heterologous expression systems. For microalgal hosts, TALEN (Transcription activator-like effector nucleases) [128] could be a feasible genome editing approach. TALEN introduces double strand breaks at a specific locus and causes the host to self-repair the damage. If a DNA sequence with a strong homology to the sequence around the cutting site is provided, the HR mechanism in cells works and use the provided DNA sequence as a template to repair the break, which achieves the accurate replacement of DNA. Serif [129] has established a TALEN-mediated gene knockout protocol for the model diatom *Phaeodactylum tricornutum*, which might be a reliable approach for genetic engineering in microalgal hosts.

## 4. Conclusions

The marine environment covers millions of phylogenetically distinct microorganisms, providing a treasure trove of biochemical diversity. The marine environment has unique characteristics compared with other terrestrial ecosystems, which are often reflected in the physicochemical properties and bioactivity of MNPs [130]. However, the biosynthetic pathways of most MNPs remain underexplored. Moreover, many marine strains have low or no expression of BGCs on usual standard media, resulting in low natural yields or the inability to synthesize MNPs [131,132,133]. An effective way to mine potential MNPs is through the heterologous expression of BGCs, expressing exogenous target BGCs in characterized hosts, and using advanced genetic engineering techniques to regulate this process [13,134,135]. With advances in DNA sequencing technology and bioinformatics tools, researchers have identified a huge potential for untapped MNPs biosynthesis. Compared to terrestrial compounds, the development of MNPs biosynthesis in heterologous hosts has barely started. Natural products from land have undergone biosynthesis with heterologous expression, with 40 from fungi alone [21], whereas only 46 BGCs from marine bacteria, fungi, cyanobacteria, microalgae, and marine metagenomes have been studied with heterologous expression as mentioned in this review. Therefore, the heterologous expression of cryptic or unknown MNPs BGCs is likely to be the next research front. The biggest obstacle for exploiting MNPs is how to mimic the culture environment, such as the adjustment of the pH, salt concentration, and composition at seawater level, let alone microbes growing near hydrothermal vents and the deep trench. Among microorganisms, it is harder for a fungal host to coexpress the entire BGC, which requires the coordination of multiple promoters working at the appropriate time [21]. Moreover, genetic manipulations are relatively intractable in more complex fungal hosts. However, generally, fungal BGCs are more likely to heterologously express in the fungal hosts. The above problems, therefore, might combine to cause difficulties for the heterologous expression of marine-derive fungal BGCs.

Efficient heterologous hosts and precise genetic manipulations are critical challenges for the heterologous expression of marine-derived BGCs. Perhaps in the future, the genetic tools of terrestrial hosts can be extended to marine microorganisms, and the technologies of genetic manipulations may make rapid progress, greatly reducing the cost and time of experiments. In addition, the establishment of various transcriptional regulatory factors, including promoters, ribosome binding sites, and terminators, will expand the toolbox of gene manipulation, allowing researchers to redesign and assemble BGCs and even complete the artificial biosynthesis of pathways. Finally, the comprehensive application of multiomics, including genomics, transcriptomics, proteomics, and metabolomics, can be integrated in heterologous expression platforms to overcome the difficulties of the industrial production of MNPs through a systematic strategy.

## Figures and Tables

**Figure 1 marinedrugs-20-00341-f001:**
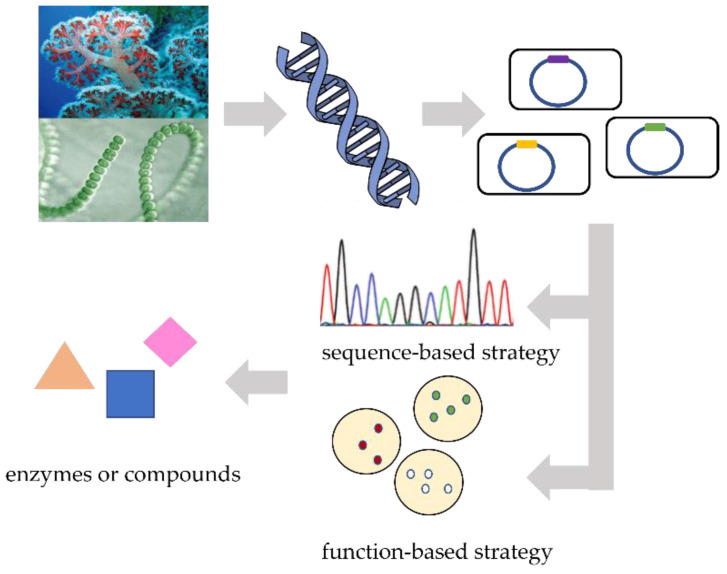
The main procedures of utilizing metagenomics in producing MNPs. Researchers first isolate metagenomic DNA from environmental samples. Then, a metagenomic library constituted by multiple cloning vectors is constructed to carry the isolated DNA, and the vectors are transferred into the host strains to express certain genes. Two strategies are used to search for genes encoding MNPs with bioactivity. A sequence-based strategy uses next-generation sequencing technology to search for homologous sequences of conserved biosynthetic genes in DNA pools, which has great advantages in mining well-studied gene classes from metagenomic DNA. A function-based strategy primarily identifies phenotypic traits of DNA library clones and screen for enzymatic activity or other products with bioactivity.

**Figure 2 marinedrugs-20-00341-f002:**
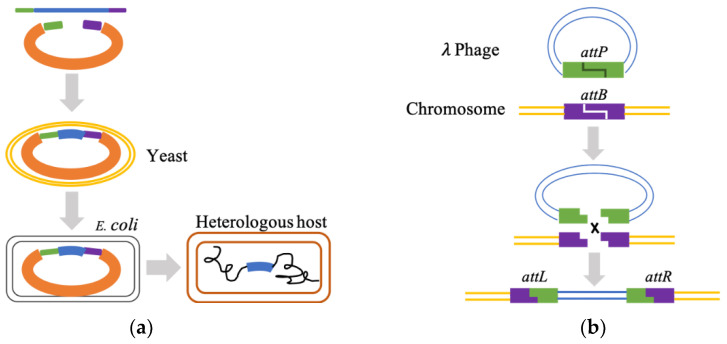
The machinery of two most frequently used in vivo BGC cloning methods (**a**,**b**) and two in vitro DNA assembly methods with complex principals (**c**,**d**). (**a**) Homologous recombination in yeast; (**b**) phage λ recombination; (**c**) overlap extension PCR method; (**d**) Gibson assembly. (**a**,**b**) are in vivo cloning methods that exploit the natural capability of DNA recombination in yeast and phage. (**c**,**d**) are in vitro cloning methods that rely on primers or exonucleases.

**Table 1 marinedrugs-20-00341-t001:** Examples of biosynthesis of MNPs in heterologous hosts.

Heterologous Host	Natural Product	MNP Type	BGC Source	Titer	Refs
**Gram-positive bacteria**					
*Streptomyces venezuelae* DHS 2001	4-*O*-demethylbarbamide	Fatty amide	*Moorena producens*	<1 μg/L	[41]
*S. venezuelae* JND2	Marineosin	Alkaloid	*Streptomyces* sp. CNQ-617	5 mg/L	[42]
*Streptomyces coelicolor* M1152	Indolocarbazole alkaloids	Alkaloid	*Streptomyces sanyensis* FMA	-	[43]
*S. coelicolor* M1152	Kendomycin B	Polyketide	*Verrucosispora* sp. SCSIO 07399	-	[44]
*S. coelicolor* M1152	RES-701-3, -4	Lasso peptides	*Streptomyces caniferus CA-271066*	-	[45]
*S. coelicolor* M1154	Lobophorins	Macrolides	*Streptomyces pactum* SCSIO 02999	-	[46]
*S. coelicolor* M1146 *Streptomyces* *lividans* TK23	Enterocin	Polyketide	*Salinispora pacifica* CNT-150	-	[47]
*S. coelicolor* M1146	Taromycin A	Lipopeptide	*Saccharomonospora* sp. CNQ-490	1 mg/L	[48]
*S. coelicolor* M1146	Streptophenazines	Pyrazines	*Streptomyces sp.* CNB-091	over 5 mg/6 L	[49]
*S. coelicolor* YF11	Fluostatin L Difluostatin A	Aromatic polyketides	*Micromonospora rosaria* SCSIO N160	-	[50]
*Streptomyces albus* De114	Nybomycin	Alkaloid	*S. albus* subsp. *chlorinus* NRRL B-24108	0.1 mg/30 mL	[51]
*S. albus* De114	Albucidin	Nucleoside derivative	*S. albus* subsp. *chlorinus* NRRL B-24108	0.4 mg/L	[52]
*S. albus*	Napyradiomycins	Terpenoids	*Streptomyces sp.* CNQ-525	-	[53]
*S. albus* J1074 *S.* *lividans* TK21	Thiocoraline	Thiodepsipeptide	*Micromonospora* sp. ML1	-	[54]
*S. albus* J1074	Berninamycins J and K	Thiopeptides	*Streptomyces* sp. SCSIO 11878	-	[35]
*Streptomyces tropica CNB-4401*	Thiolactomycin	Polyketide	*Streptomyces. pacifica*	3-fold higher	[55]
*Bacillus subtilis*	Preamicoumacins	Isocoumarin	*Bacillus subtilis 1779*	-	[56]
**Gram-negative bacteria**					
*Anabaena* sp. PCC 7120	Cryptomaldamide	Hybrid tripeptide	*Moorena producens* JHB	25.9 ±3.6 mg/L	[57]
*Anabaena* sp. PCC 7120	Lyngbyatoxin A	Terpenoid indole alkaloid	*Moorena producens*	3.2 mg/L	[58]
*Anabaena* sp. PCC 7120	Pendolmycin	Indolactam alkaloid	*Marinactinospora thermotolerans* SCSIO 00652	-	[59]
*Anabaena* sp. PCC 7120	Teleocidin B-4	Indolactam alkaloid	*Streptomyces blastmyceticus* NBRC 12747	-	[59]
*Escherichia coli*	Patellamides A and C	Cyclic peptides	*Prochloron* spp.	-	[60]
*E. coli*	Xylanase	Protein	*Marinimicrobium* sp. LS-A18	-	[61]
*E. coli*	α-amylase	Protein	*Zunongwangia profunda* (MCCC 1A01486)	-	[62]
*E. coli*	Kappa -Carrageenase	Protein	*Zobellia* sp. ZM-2.	9-fold higher	[63]
*E. coli*	Chitinase *Pt*Chi19p	Protein	*Pseudoalteromonas tunicata* CCUG 44952T	-	[64]
*E. coli*	Lyngbyatoxin A Indolactam-V	Indole alkaloids	*Moorena producens*	25.6 mg/L 150 mg/L	[65]
*E. coli*	Surfactant	Lipopeptide	*Bacillus licheniformis* NIOT-06	-	[66]
*E. coli*	Alginate lyase	Protein	*Vibrio* sp. QY102	0.58 g/L	[67]
*E. coli*	Alterochromide	Lipopeptide	*Pseudoalteromonas piscicida* JCM 20779	-	[68]
*E. coli* BL_21_(DE_3_)	Thalassomonasins A and B	Lanthipeptides	*Thalassomonas actiniarum*	1.9 mg/L	[69]
*E. coli* BL_21_(DE_3_)	Marinomonasin	Tricyclic peptide	*Marinomonas fungiae*	-	[70]
*E. coli* BL_21_(DE_3_)	Alkaline lipase	Protein	Marine sponge metagenome	-	[71]
*E. coli* BL_21_(DE_3_)	Antifungal peptide	Peptide	Seawater metagenome	-	[72]
*E. coli*	Vibrioferrin	Tricarboxylic acid	Tidal-flat sediment metagenome	92.6 mg/L	[27]
*E. coli*	Avaroferrin Putrebactin	Alkaloids	Deep-sea metagenome	11.5 mg/L 1.2 mg/L	[73]
*E. coli*	Bisucaberin	Hydroxamate	Deep-sea metagenome	8.4 mg/L	[74]
*E. coli*	Lipase	Protein	Marine sponge metagenome	-	[75]
*E. coli*	Halichrome A	Biindole	Marine sponge metagenome	-	[76]
*E. coli*	Esterase	Protein	Marine mud metagenome	-	[77]
*E. coli*	Chitosanase	Protein	Marine mud metagenome	-	[78]
*E. coli*	Laccase	Protein	Marine microbial metagenome	-	[79]
*E. coli*	Desferrioxamine E Desferrioxamine D2 Desferrioxamine X1 Desferrioxamine X2	Siderophores	Fusion of marine metegenomic DNA and a terreastial bacterium	27 mg/L 53 mg/L 7.1 mg/L 1.2 mg/L	[80]
*Myxococcus xanthus*	Haliangicin	Polyketide	*Haliangium ochraceum* SMP-2	10-fold higher	[81]
**Fungi**					
*Aspergillus nidulans* RJMP1.59	Beauvericin	Cyclic lipopeptides	*Fusarium proliferatum* LF061	668.97 mg/L	[82]
**Microalgae**					
*Fistulifera solaris*	Prostaglandins	Fatty acids	*Agarophyton vermiculophyllum*	1290.4 ng/g cell dry weight	[83]

**Table 2 marinedrugs-20-00341-t002:** Summary of BGC cloning methods applied in the heterologous expression of MNP BGCs.

BGC Cloning Methods	Advantages	Disadvantages	Refs
in vivo			
TAR	Directly clone MNP BGCs up to 300 kb	False positives	[45,47,48,49,56,57,58,68]
λ/Red	Direct modification of DNA within *E. coli* and this method is independent of restriction sites	The efficiency drops sharply as the size of the cassette increases.	[43,48,59]
Red/ET	A powerful tool for DNA subcloning and DNA modifications	Hard to mediate homologous recombination between two linear DNA	[41,42]
SIRA	Efficient genomic assembly of large MNP BGCs (>100 kb)	Requirement of specific sites integrated into the chromosome	[48,55]
in vitro			
Cosmid library	Simple construction of cosmid library	Tedious screening	[35,43,50,53,54,81]
Fosmid library	More stable than the conventional cosmids; allows for both low/single copy number and high copy propagation	A fosmid vector only accepts small BGCs (up to 45 kb).	[27,60,65,73,74,75,76,77,78]
Restriction enzyme-mediated	High efficiency; simple operation	Strict limitations on the restriction sites in the target sequences	[61,62,63,67,69,70,71,72,79,80]
BAC	Clone large-sized DNA fragments from the complex genome	Tedious screening	[44,46,51,52]
Overlap extension PCR	No need for restriction endonucleases or T4 DNA ligase	Possibility of introducing mutations	[45]
Gibson assembly	No concerns for internal restriction enzyme cutting sites	Not applicable for the DNA fragments with high GC content (over 60%)	[82]

TAR, transformation-associated recombination; SIRA, serine integrase recombination assembly; BAC, bacterial artificial chromosome.

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
