# Peer review of "Recent Advances in the Heterologous Expression of Biosynthetic Gene Clusters for Marine Natural Products"

_marinedrugs, 2022, doi:10.3390/md20060341_

Round 1
Reviewer 1 Report
The authors describe the ways to unlock silent biosynthetic gene clusters in microorganisms. One of the ways described in this report is heterologous expression. The authors describe the up to date progress in this area. A very minor comment, can the authors explain in more details the figure caption for figures 1 and 2.
Reviewer 2 Report
The review gives a good overview of advances in heterologous expression of synthetic gene cluster for marine natural products, being the last frontiers of drug discovery. Table 1 have to be format in horizontal view order to be more readable. Genetic engineering techniques are explored but CRISPR Cas9 and Talen approach are totally missing together with microalgae as host organisms. It would be possible add few sentences on the topic or a new paragraph updated.
Reviewer 3 Report
The literature on marine natural compounds has become extensive and includes a sufficient number of the most recent reviews on the subject (biosynthesis, sources, diversity, etc.). The review "Recent advances in the heterologous expression of biosynthetic gene clusters for marine natural products" submitted by Yushan Xu et al. to the journal Marine Drugs aims to fill a gap in the analysis of heterologous expression data of biosynthetic gene clusters from marine microorganisms. It undoubtedly meets the needs of the scientific community. The strength of this review are well-structured tables.
Unfortunately, text is not easy to read. There are a lot of inaccuracies and wrong references.
There are some important notes:
Ref. 1. This overview article does not support the information provided in this proposal (lines 26-28).
Ref. 3. This article cannot be used as a reference for these proposals (lines 30-33). It reports “the identification of the gene cluster encoding production of the structurally unique antibiotic nybomycin”.
Ref. 6. Please, replace it with link to FDA.
Ref. 7. This overview article does not support the information provided in this proposal (lines 39-41).
Ref. 9. In this overview article questions related to chemical synthesis as “wholly chemosynthesis is challenging” were not raised or considered at all. This proposal (lines 48-50) does not support by the ref. 9.
Ref. 10. This proposal (lines 51-52) does not support by the ref. 10, because it is about platforms using gene clusters not on genetic basis. There should be another link related to the description of the structure of biosynthetic gene clusters.
Ref. 12. This overview article does not support this proposal (lines 58-59). Kalaitzis et al. did not say about “Genome mining” and “the potential functions of metabolites”. Rephrase it, please. In concluding remarks they said “Although genetics-based methods alone may not guarantee success, they certainly provide useful information that was not previously available thus providing a competitive advantage for the discovery of novel microbial natural products”.
Ref. 13. Please, provide this proposal by review articles or research articles.
Ref. 19. In this reference, the phrase "strong promoters" appears once, in the introduction. Provide links to reviews or a set of articles describing such examples.
Ref. 22. It says about “mixed fermentations can stimulate the production of secondary metabolites” in laboratory conditions. Please, provide lines 90-91 by correct reference(s).
Ref. 23. One link is not enough to support this claim (lines 95-96, 122-123).
Ref. 25. One link (Meng et al. Developing fungal heterologous expression platforms…) is not enough to support this claim (lines 116-119).
Ref. 73. This link is not available via a google search.
Ref. 74. One link is not enough to support this claim (lines 164-165).
Line 189. Why is a reference 47 here?
Line 211. One link (ref. 3) is not enough to support this claim.
Lines 372-374. The reference 106 does not support this claim.
Line 375-377. One link (ref. 107) is not enough to support this claim. Provide it with “fresh” references.
Line 377-380. Provide these proposals with more references.
Here are minor remarks.
Table 1:
Should be Kendomycin B.
Should be Patellamides (or A and C).
Should be kappa-carrageenase.
Should be Thalassomonasins A and B.
Should be lipase.
Should be chitosanase (not chitinase!)
Line 65. Please, decipher the abbreviation “CMNPD”.
Line 186. Should be “Lyngbya”.
Lines 196, 198, 202 and 204. Please, use PCC 7120.
In the text, there are unsuccessful terms and expressions that change their meaning:
Lines 41-42. What does it mean “radiofluorescence microscopy”?
Line 105. Specify meaning of “recombination” in this context.
Line 273. “Long DNA aggregates”.
There is not a link to Figure 1 (lines 128-131).
Reviewer 4 Report
The review "Recent advances in the heterologous expression of biosynthetic 2 gene clusters for marine natural products" focus on Marine natural products (MNP) as active metabolites for therapeutic agents. Due to weak or no expression of many biosynthetic gene clusters (BGCs) under at least laboratory conditions the authors describe heterologous expression systems as solution. Therefore the current challenges are described and progress summarized.
Abstract:
- The phrase "aforementioned challenges" should be replaced or the challenges should be more clearly mentioned. So far the abstract is explaining challenges in homologous systems but not the exact problems/challenges for the heterologous expression systems.
Introduction:
- It would be good to add a table or more examples except of Brentuximab vedotin to show the importance of MNPs and versatile applications.
- In line 47 it is written medicinal. Does the authors mean medical?
- In Line 52 one whitespace too much after Today,
- ONly be careful with the wording database and bioinformatics tool. antiSMASH is called in the same sentence both but is a tool.
- Also the authors should state some more databases and tools as they are claiming a wide variety of them. MAybe a table for naming instead of refering to another review in the sentence.
- The OSMAC strategy is quite prominent in the introduction without stating the challenges and problems in general. So it would be good to get first an overview of methods for the homologous expression systems and the still existing problems.
- Also as teh review wants to focus on heterologous expression systems the clear separation from OSMAC and other tools and the upcoming challenges by the heterologous systems.
- I did not find the usage of Figure 1 as reference in the main text (Figure 1).
- Also as the authors here mention it as the two main metagenomic methods it is not clearly described in the main text and for this is now used in the heterologous expression systems. In the end of the introduction the authors state the two main steps for the expression systems but did not introduce problems or examples here.
- Table 1 is referenced two times at different positions one time as specific for actinomycetes and cyanobacteria and the other time for marine BGCs. So it would be good to specify it more or described if it is at stage only possible for actinomycetes and cyanobacteria or only marine BGCs can be analyzed.
Heterologous hosts for MNPs
- The phrase "the greater the transcription factors works" are uncommon for me. Did the authors mean the higher is the chance that the transcription factors will work? Or is it an linear increase in the efficacy?
- Again Table 1 is named now including all? It should be considered to clarify by making more than one big table and divide in based on hosts or MNPs or BGCs.
- An explanation or more description of codon usage pattern and evolution of similar hosts would be also good for the description. LAter the authors claim E.coli is a good host of bacteria but it is evolutionary distant from other bacteria so what is the more important role. Well known model organism or evolutionary close host?
- It should before mentioning the four different types of organisms clearly described in the introduction when are they used for what. So give the researcher some classes or separation hints.
Genetic manipulations for BGCs
- The beginning of this paragraph is not giving clear advanced in the BGC methods. The authors write about disadvantages and advantages in the Table but it is not very good classified. So it is hard to udnerstnad in the first place when in vitro assembly or in vivo cloning and what is now the general benefit of each of them.
- Why are in the Table 2 more methods than then visualized in the "common" BGC cloning methods? How are the authros discriminating between "common" and "uncommon"?
- The chapter should be also splitted maybe in some more parts like before the hosts to clarify the different methods and show specific problems and current solutions for the single methods or classes of methods.
- Also some qunatitative measures are not really defined. Talking about Low positive rate or high GC content is very objective. When is a GC content high? What is a low positive rate and compared to what other methods?
Conclusion:
- Is this a fact "Compared to terrestrial compounds, the development of MNP biosynthesis has barely started, and the heterologous expression of cryptic or unknown MNP BGCs is likely to be the next research front." or more an hypothesis.
- Good would be to get an comparison then of numbers for terrestrial and marine compounds.
- What are the problems to mimic marine conditions?
- starting points for new heterologous expression systems or other problems and challenges should be here named again to clearify obstacles to overcome.
